

# Divergent effects of short-term and continuous anthropogenic noise exposure on Western Bluebird parental care behavior

Kerstin Ozkan[1], Jordan M. Langley[1], Jason W. Talbott[1], Nathan J. Kleist[2] and Clinton D. Francis[1]

[1] Department of Biological Sciences, California Polytechnic State University—San Luis Obispo, San Luis Obispo, California, United States
[2] Department of Ecology and Evolutionary Biology, University of Colorado at Boulder, Boulder, Colorado, United States

## ABSTRACT

Sensory environments are rapidly changing due to increased human activity in urban and non-urban areas alike. For instance, natural and anthropogenic sounds can interfere with parent-offspring communication and mask cues reflective of predation risk, resulting in elevated vigilance at the cost of provisioning. Here we present data from two separate studies involving anthropogenic noise and nestling provisioning behavior in Western Bluebirds (*Sialia mexicana*): one in response to short-term (1 h) experimental noise playback and a second in the context of nests located along a gradient of exposure to continuous noise. In the short-term playback experiment, nests were sequentially exposed to trials with either traffic noise or a silent audio track. The effect of the playback type interacted with the effect of the order in which trials were presented. The outcome was that provisioning rates during second trials with the silent track playback were higher than provisioning rates during noise playback on first or second trials, but not first trials with the silent track playback. Additionally, failed provisioning attempts only occurred during noise trials. In contrast, provisioning rates increased with the amplitude of noise among nests located in a gradient of continuous noise exposure. For nests along the noise gradient, the latency to resume provisioning behavior following human disturbance from approaching the nest negatively covaried with noise exposure amplitude. Specifically, birds resumed provisioning behavior more quickly with increased noise amplitude. Collectively, both studies demonstrate that noise can influence avian parental care of offspring, but the direction of the effect of noise are opposite. This difference could reflect variation in populations, noise characteristics or latent environmental contexts, or different ages of nestlings. However, it is also possible that the divergent responses reflect important differences in organismal responses to short-term *versus* long-term noise exposure. The possibility of mismatches in responses to short-term *versus* long-term noise exposure should be the focus of additional research, especially because short-term noise exposure experiments are often used to understand the consequences of noise pollution for organisms living in noisy environments.

Corresponding author
Clinton D. Francis,
cdfranci@calpoly.edu

# INTRODUCTION

Urbanization is an omnipresent threat to biodiversity that is increasing in many parts of the world (*Dominoni et al., 2020*). As urbanization expands, so do stimuli that alter the way animals experience the world around them. Anthropogenic changes to an organism's sensory environments can create novel environmental conditions that disrupt the ability to perceive once-reliable environmental cues (*Ferraro, Le & Francis, 2020*) and have the potential to result in dramatic behavioral, reproductive, and community-level responses among wildlife (*Gaston et al., 2013*; *Shannon et al., 2016*). Despite a growing body of research, many aspects of how sensory pollutants alter species behaviors, and which life stages are most heavily impacted, remain poorly understood.

Noise can interfere with behavior by changing the way individuals perceive and interact with their surroundings through a variety of mechanisms including acoustic masking and distraction (*Dominoni et al., 2020*). Acoustic masking occurs when the frequency of a sound, such as traffic noise, the sound from wind blowing through leaves or the sound of chorusing insects, overlaps the frequency of another sound. This overlap can impair the detection or discrimination of target sounds, such as the call of a mate or the rustling sound of prey (*Dominoni et al., 2020*). The most well-studied examples of masking involve impaired communication among conspecifics (reviewed in *Francis, Phillips & Barber, 2023*). For instance, noise has been shown to impair communication between parent and offspring through acoustic masking, with nestlings exposed to noise begging less upon their parents' arrival (*Leonard & Horn, 2012*; *Lucass, Eens & Müller, 2016*). Masking by noise can also impair an individual's ability to detect an approaching predator (reviewed in *Francis, Phillips & Barber, 2023*). To compensate for the loss of their auditory sense, animals often rely on other sensory modalities, such as vision. However, unlike audition, vision is directional, and effective threat detection requires visual scanning (*Rabin, Coss & Owings, 2006*; *Meillere, Brischoux & Angelier, 2015*). This increase in visual vigilance behavior (*i.e.*, scanning) can decrease the amount of time an adult spends on other behaviors like parental care and foraging (*i.e.*, foraging-vigilance tradeoff; *Sweet et al., 2022*). Additionally, species that rely on audition to capture prey, such as owls, experience reduced foraging efficiency when exposed to traffic noise (*Mason, McClure & Barber, 2016*; *Senzaki et al., 2016*). Background noise can also impair localization of hidden prey for diurnal songbirds, such as the American Robin (*Turdus migratorius*) (*Montgomerie & Weatherhead, 1997*); thus, it is possible that anthropogenic noise could impair prey cue detection and hunting success in other diurnal birds as well. Collectively, these consequences of masking, increased vigilance, reduced foraging efficiency, and missed detections, all have the potential to interfere with critical parental care behaviors such as provisioning food for young.

Here, we studied the effects of noise exposure on parental care in Western Bluebirds (*Sialia mexicana*). Western Bluebirds are known to readily nest in areas with high noise

exposure (*Kleist et al., 2017*), but chicks from noisy nests experienced reduced hatching success, hormone dysregulation, and altered nestling growth (*Kleist et al., 2018*)—all of which could be linked to effects of noise on parental care, specifically nestling provisioning behavior. Changes in provisioning behavior have been shown in other species when exposed to disturbances, including noise exposure (*Lucass, Eens & Müller, 2016*; *Injaian, Taff & Patricelli, 2018*), and is often used to study parental care as it is easy to accurately measure.

We studied the influence of noise on nestling provisioning in two locations and noise stimuli contexts because they offered opportunities to measure different responses to noise that are associated with parental care and, when considered together, should lead to greater generalizability. First, at a nest box system in California, USA we measured provisioning during consecutive 1-h trials that included either playback of traffic noise or a silent audio track with no acoustic energy. Second, we quantified provisioning behavior at nests located along a gradient of exposure to continuous noise at a nest box system in New Mexico, USA. Because noise can be a distracting stimulus, interfere with parent-offspring communication, and impair acoustic surveillance for threats, we expect to see a change in adults' provisioning rates when exposed to noise. If noise pollution interferes with the provisioning behavior because it impairs surveillance for threats, then observational data from short-term experiments should reveal that Western Bluebirds exposed to noise will approach the nest more cautiously and spend more time outside the nest box than when exposed to control conditions and this should also result in fewer nest visits per trial. Similarly, we predict that birds will provision less with increases in continuous noise levels and that birds will return to the nest more slowly following a nest disturbance, where parents respond to the disturbance in a similar way as they would to predation (*i.e.*, risk-disturbance hypothesis; *Frid & Dill, 2002*). Because noise can also interfere with parent-offspring communication and chicks may fail to hear the arrival of their parents, we predict that there would be more failed provisioning attempts in noisy conditions for both short-term playbacks and with increases in continuous noise levels (Table 1).

# MATERIALS AND METHODS

## Short-term noise exposure study

We investigated Western Bluebirds' behavioral responses to noise pollution using field-based experiments located throughout rangelands and oak woodlands adjacent to California Polytechnic State University's campus on the Central Coast of California (35°19′18″ N 120°38′27″ W). In this area, 180 nest boxes are located 60–100 m apart on fences (*Ferraro, Le & Francis, 2020*). Occupied nest boxes were sufficiently separated in space and time to avoid the effects of playbacks on neighboring nests.

We monitored nest boxes from March to June 2022. Boxes were initially checked every 2 weeks for signs of nesting material. Once a complete nest had formed, we monitored nests every 4–7 days until the completion of a full clutch and/or signs of incubation. The expected hatch date was calculated by adding 2 weeks to the completed clutch date and/or the first sign of incubation. Nearing the expected hatch date, we checked nests daily until at

**Table 1 Consequences of acoustic masking and predictions of how these non-mutually exclusive mechanisms may impact behavior under both long and short-term noise exposure in Western Bluebirds.**

| Mechanism | Prediction |
|---|---|
| Increased vigilance | Because noise increases the perception of risk (*Meillere, Brischoux & Angelier, 2015*; *Quinn et al., 2006*), birds should approach the nest more slowly, spending more time within 10 m of the nest during the approach and provision less. Similarly, birds should provision less and have a longer latency to resume provisioning under continuous noise exposure. |
| Reduced foraging | Because increased visual vigilance in noise comes at a cost to foraging rate (*Sweet et al., 2022*) and noise can reduce foraging efficiency by masking prey sounds (*Montgomerie & Weatherhead, 1997*; *Mason, McClure & Barber, 2016*; *Senzaki et al., 2016*), there should be a decline in provisioning rate, but not necessarily time spent within 10 m of the nest box. |
| Missed detections | Because noise contributes to chicks failing to hear the arrival of parents (*Leonard & Horn, 2012*; *Lucass, Eens & Müller, 2016*), there should be more failed provisioning attempts with short-term experimental noise exposure and/or increased sound levels. |

least one egg had hatched (day 0) and left them undisturbed until experiments took place when nestlings were five days old (day 5).

## Experimental treatment

We conducted a repeated-measures playback experiment on 24 unique nests with 5-d old nestlings with and without traffic noise exposure. The order of exposure to either the silent audio track or one of six traffic noise tracks was randomly assigned to balance potential variation in provisioning rates throughout the day and to control for order effects. Although previous studies observed almost immediate changes in behavior after changes in acoustic conditions (*e.g.*, *Gross, Pasinelli & Kunc, 2010*; *Shannon et al., 2014*), traffic or silent tracks were broadcast for 15-min prior to the beginning of any trial period (*i.e.*, pre-trial broadcast) to ensure that behavioral changes were reflective of the acoustic environment rather than a sudden change in sound levels with treatment (*Le et al., 2019*). In addition, a 15-min rest period was included between each trial. Observation periods for each trial lasted 60 min (Fig. 1C).

Nests were exposed to one of six traffic noise recordings from different locations on local roadways (*e.g.*, *Ferraro, Le & Francis, 2020*). Multiple recordings were used to increase the generalizability of any potential responses and to minimize the influence of any acoustic characteristics that may have been unique to one stimulus. Nevertheless, all recordings were made from busy roadways and power spectra reveal a similar distribution of energy across frequencies (*Mulholland et al., 2018*). Ambient conditions consisted of the playback of a silent track with no acoustic energy to control for possible influences of electromagnetic noise (*Engels et al., 2014*). Acoustic stimuli were broadcast from an Octasound 800 speaker with a Lepai LP-2020TI amplifier and an Olympus LS-P2 audio player (*e.g.*, *Le et al., 2019*; *Reed et al., 2021*) that were placed 10–15 m from the nest such that received levels of the traffic noise playback averaged at 65 dB(A) at the nest (2-min time-integrated [Leq] sound level, re 20 µPa). Received sound levels were measured using a Larson Davis 824 at the entrance of the nest box for 2 min following the 2-min gradual increase in amplitude at the beginning of the pre-trial broadcast (Fig. 1C).

We observed bird activity from an observation blind placed 25–30 m from the nest box (Fig. 1A). A voice recorder was used to document provisioning visits made by parents and

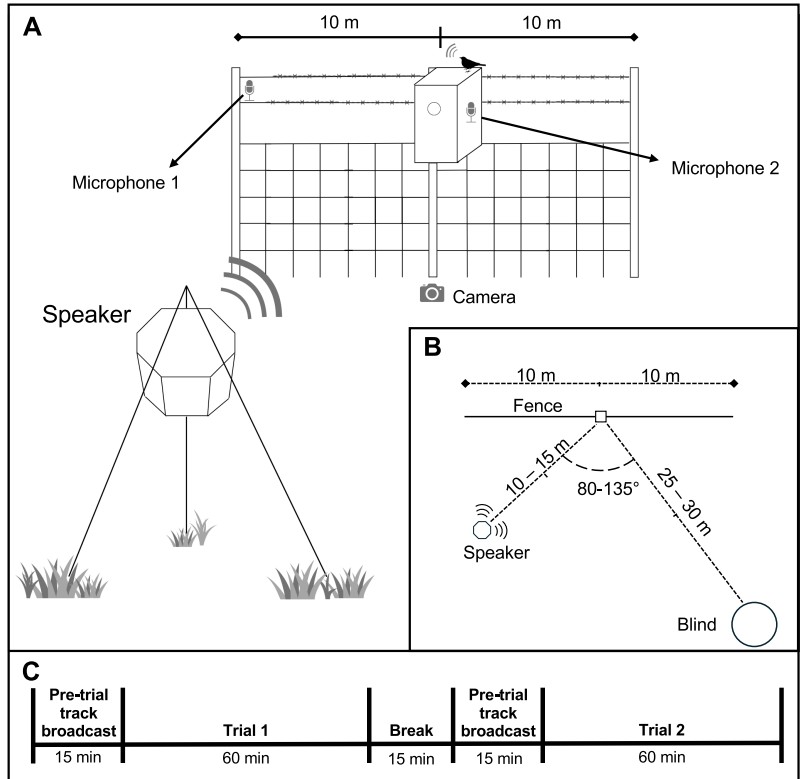

**Figure 1 Layout and timeline of short-term experimental noise playback experiment.** (A) Overview of the short-term experimental noise playback design. The speaker that broadcast traffic noise or the silent track was placed 10–15 m from the nest. Microphone 1 denotes the microphone placed on the fenceline to record parent vocalizations when near the nest. Microphone 2 was placed inside the nest box to record chick begging calls and parent contact calls. A camera was also placed directly below the nest facing up to help verify the time of provisioning events and identity of parents. Ten meter distances along fence reflect radii of perch distances for quantifying time within 10 m from the nest. (B) Birds eye view of experimental setup displaying distances of equipment and observation blind from the nest box. (C) Timeline of experimental trials. Each nest received two trials: one with a traffic noise playback and another with a silent audio track. The order of each was randomized. Prior to trials, the audio tracks were broadcast from the speaker for 15 min. When the broadcast was a traffic noise trial, the amplitude was gradually increased over a 2-min period before reaching a time-averaged 65 dB(A) exposure amplitude at the nest, which was set using a Larson-Davis 824 Sound Level Meter.

the time spent outside the nest box within a 10-m radii. Small pieces of flagging tape were placed on the fence line at 10 m to help assess distance categories. Two video cameras (Canon Vixia HF R50) were used to verify the timing and identity of individuals and food items: one camera was mounted near the ground facing up at the nest box and a second camera was used in the blind. Additionally, two acoustic recorders were used to document adult contact calls and chick begging calls. One Roland R05 recorded chick begging calls with an Olympus ME-15 microphone (100–12,000 Hz) and custom windscreen taped to the back-right interior of the nest box lid. The second R05 recorded adult contact calls at a distance of 10 m from the nest box with a MicW iShotgun microphone oriented toward the nest and away from the playback speaker to minimize noise in the recording (Fig. 1). Once all recording equipment was set up, a Wrentit

(*Chamaea fasciata*) call was broadcast to signal the start time of the experiment and to ensure that analyses occurred at the appropriate start time. We used a Wrentit call because it is a common species throughout the area and is not considered a competitor and thus should not elicit a response by Western Bluebirds.

Behavioral responses were recorded as video files and audio recordings from both the observation blind and directly below the nest box. In practice, videos were used to confirm observations noted in the voice recorder from observers in the blind. For instance, videos were used to verify successful or failed provisioning attempts and to precisely record the duration time an individual spent within 10 m of the box. Consultation of videos to confirm observations were scored blind to treatment. We defined provisioning behavior as when a Western Bluebird was observed to enter the nest box with food (*e.g.*, insects or berries) and later exited the nest box without food in its bill. We defined a failed provisioning attempt as when a bird entered the nest box with a food item and exited the box with the food remaining in its bill or when a bird was perched or directly hovering at the nest box entrance with food but did not enter the nest box to feed its young. This research was approved by the California Polytechnic State University Institutional Animal Care and Use Committee (protocol 2105).

## Continuous noise exposure study description

This part of the study took place in 2012 and 2013 at a nest box system in northwestern New Mexico (36°55′39″ N 107°41′50″ W) where 240 nest boxes were distributed across 12 pairs of noise treatment and quiet control sites (10 boxes per site) (*Kleist et al., 2017*, *2018*). Noisy and quiet sites were geographically paired to help control for spatial autocorrelation in latent environmental conditions (see *Kleist et al., 2017*, *2018* for details). Nest boxes were systematically arranged surrounding sites between 75 and 175 m from the center of gas well pads. On noisy sites large compressor engines and fans produced high-amplitude, low-frequency noise continuously throughout the entire nesting cycle. Doing so achieved a gradient of noise exposure among nest boxes from gas well compressors on noisy sites (see *Kleist et al., 2017* for details). In this system only a small fraction of nest boxes are occupied by nesting songbirds in a given year and we observed provisioning behavior in subset of nest boxes ($n = 13$) used by Western Bluebirds during the 2012 and 2013 breeding seasons.

Nests were monitored for activity from May to July each year from nest discovery until they fledged or failed. When chicks were 12 (± 1) days old, we installed a video camera (Kodak Playsport zx3) approximately 4 m from the nest box and recorded nestling provisioning for approximately 2 h (mean = 111, SD = 14 min, range = 86 to 132). We started the recording immediately after installing the video camera to measure the latency to resume provisioning behavior. We also calculated time-averaged sound pressure levels (*i.e.*, $L_{eq}$ in unweighted decibels (dB), fast response, re. 20 µPa) for each nest from 1 min measurements with a Larson-Davis 824 because shorter interval measurements from separate days, times and conditions were found to be highly repeatable in this system (*Kleist et al., 2018*).

Similar to the short-term experiment, we scored successful and failed provisioning events from videos, but did not separate these by parental sex. Additionally, because

nesting adults respond to the approach to the nest by a researcher in a manner similar to real predation events (*i.e.*, risk-disturbance hypothesis; *Frid & Dill, 2002*), we calculated provisioning rate as the number of visits per hour, but quantified over the duration of the observation period after provisioning resumed (*i.e.*, latency to provision following nest disturbance). Thus, provisioning rates were calculated over a mean of 104 (SD = 14 min). This work was approved by the University of Colorado Boulder Institutional Animal Care and Use Committee (Protocol 1404.03). Although use of audio recorders or cameras during both the short-term experiments and at nests exposed to continuous noise could potentially impact the privacy of people by inadvertently recording them, this was not an issue in this study. We did not complete short-term trials when people were present and could be inadvertently recorded. In the continuous noise study, our nest boxes are in a very remote area where we have never encountered people near our nests since starting to work in this system in 2005.

## Analyses

*Short-term noise exposure.* Twenty-three nest boxes were used for analysis. All statistical tests were performed using R Version 4.2.0.

For analyses of the number of provisioning events per trial, we built generalized linear mixed models (GLMMs) with Poisson error with the glmer function in the lme4 R package (version 1.1–31). Using the glmer.nb function in the lme4 R package, we built negative binomial GLMMs models for the number of failed provisioning attempts. A linear mixed effects model (LMM) was created with the lmer function in the lme4 R package for analysis of the time an adult spent <10 m from the nest box. Nest was treated as a random effect for all analyses to account for the repeated measures design and non-independence of data. Sound file was also initially treated as a random effect to account for multiple stimulus files but was removed from the model as the variance was <0.0001 (*Bates et al., 2015*).

Fixed effects for provisioning and failed provisioning attempt models included trial treatment (noise/ambient), order of treatment, brood size, ordinal date and the amount of time the parent spent within 10 m of the nest box. We also included an interaction between treatment and trial order (*i.e.*, first or second). For the models explaining the amount of time the parent spent within 10 m, we transformed the response by taking the natural logarithm of time spent with 10 m +1 and included trial treatment (noise/ambient), order of treatment, brood size, ordinal date and the interaction between treatment and trial order. Order in which the treatments occurred was included to investigate potential order effects influencing behavior. Brood size was included to account for variation in brood size among the nests, which also has the potential to impact provisioning behavior. In these models ordinal date was included to account for seasonal variation in provisioning behavior. The amount of time parent spent within 10 m of the nest box was included to examine whether noise influenced a parent's hesitancy to approach the nest box, thus impacting the number of provisioning events. We tested for an interaction between treatment and trial order to determine whether provisioning behavior within each treatment varied by trial order. We centered and scaled ordinal date and time a parent spent within 10 m to improve model convergence.

We verified that models met model assumptions by inspecting residuals using the simulateResiduals function in the DHARMa R package (version 0.4.6) and by verifying model dispersion was near 1 using the dispersion_glmer function in the blmeco R package (version 1.4). For failed provisioning attempts, we checked the model using the check_zeroinflation command in the performance R package (version 0.10.2). Using the dredge function in the MuMIn package in R (version 1.47.1), we compared models with Akaike's Information Criterion corrected for small sample size ($AIC_c$) and considered models with $\Delta AIC_c \leq 2$ as strongly supported. We calculated marginal effects using the ggeffects R package (version 1.2.2). We used the estimated effect size and 95% confidence intervals (95% CIs) to interpret the magnitude and precision of model predictor estimates. When parameter estimates appear in multiple supported models, we present estimates from the highest-ranked model.

*Continuous noise exposure.* We initially modeled provisioning rate, failed provisioning rate and the latency to resume provisioning using linear mixed-effect models with the lmer function in the lme4 R package. We treated sound amplitude, lay date, clutch size and time of day as fixed effects and site as a random intercept. However, in all cases, we removed the random effect of site because it explained near-zero variance (*i.e.*, <0.0001) and refit models as linear models with the lm function in R (*Bates et al., 2015*). We ranked models with $AIC_c$ and used effect sizes and 95% confidence intervals to gauge the size and precision of the influence of predictors. We used model diagnostics to verify that model assumptions were met for all strongly supported models. Cook's distance identified a single record with potentially high leverage for the provisioning rate and latency to resume provisioning models. We reran supported models without the records with potentially high leverage and found that their exclusion did not alter the interpretation of the results. Thus we present the results with their inclusion below.

## RESULTS

### Short-term noise exposure

We performed a total of 48 experimental trials on 24 individual nesting attempts, with each receiving a control and noise treatment. Three models were competitive following model selection (*i.e.*, $\Delta AIC_c \leq 2$). The top model explaining provisioning behavior included parental sex, treatment, trial, time the adults spent <10 m from the nest, and the interaction between treatment * trial order (Table 2). The second-ranked model included the same variables as the top model, plus brood size. The third-ranked model included parental sex, trial order, and time the adults spent <10 m from the nest.

Based on marginal effect estimates from the highest-ranked model, among first trials, treatment alone did not have a strong influence on provisioning behavior. However, the order in which the treatment occurred had a strong effect on the provisioning behavior of Western Bluebirds. During second trials, parents typically provisioned nestlings only 4.17 times per hour (95% CI [3.15, 5.52]) when exposed to noise, but 6.81 times per hour (95% CI [5.30, 8.75]) under ambient sound conditions (Fig. 2). Provisioning rates of 6.81 times per hour during second trials in ambient sound conditions were also higher than provisioning rates when exposed to noise during first trials (predicted = 4.97/h, 95% CI

**Table 2 Model selection table for variables explaining provisioning behavior in response to short-term traffic noise playback.**

| Model | K | AIC$_c$ | ΔAIC$_c$ | weight |
|---|---|---|---|---|
| **Parent sex** (+), treatment (+), **trial** (+), **time within 10 m** (+), **treatment*trial** (+) | 7 | 460.46 | 0.00 | 0.42 |
| Brood size (+), **parent sex** (+), treatment (+), **trial** (+), **time within 10 m** (+), **treatment*trial** (+) | 8 | 461.62 | 1.16 | 0.23 |
| **Parent sex** (+), **trial** (+), time within 10 m (+) | 5 | 462.41 | 1.95 | 0.16 |
| Null | 2 | 467.27 | 6.81 | 0.00 |

Note:
   Time within 10 m was centered and scaled. +/− shows the direction of the trends. Bolded values show parameters with 95% CIs that do not overlap zero.

[3.81, 6.49]). For ambient sound condition trials, provisioning rates were 71% higher in second relative to first trials. The pattern was reversed among noise exposure trials, where provisioning rates in second trials were 16% lower than first trials, but the precision of the estimated difference was low (Table 3). Among other influential predictor variables, parental sex had a strong effect on provisioning rates, with males provisioning less than females (Fig. 2 and Table 3). Additionally, adults that spent more time <10 m of the nest box provisioned their nestlings more frequently ($\beta = 0.12 \pm 0.05$, 95% CI [0.01, 0.22]; Fig. 2 and Table 3).

Although the top-ranked model explaining parent time <10 m from the nest box included only the random effect of nest ID, other competitive models included treatment, trial or the interaction, respectively (Table 4). From the second-ranked model, during first trials parents spent less time <10 m during noise trials ($\beta = -0.99 \pm 0.48$, 95% CI [−1.95, −0.02]), but more time <10 m of the nest box during second trials that included noise ($\beta_{\text{treatment*trial}} = 2.02 \pm 0.79$, 95% CI [0.40, 3.64]). However, because the top-ranked models was the null, interpretation of these treatment and trial order effects should be interpreted with caution (Table 5).

Analysis of failed provisioning attempts produced 11 candidate models, none of which included the null (ΔAIC$_c$ 26.40). All top models included parental sex and treatment as predictor variables. Trial order was in six models, two of which also included the interaction with treatment in 15 of the 25 models (Table 6). The top model included parental sex, treatment and brood size. Based on this model, the failed provisioning rate in noise trials was over six times higher (predicted = 0.19/h, 95% CI [0.06, 0.55]) than control trials (predicted = 0.03/h, 95% CI [0.01, 0.13]; $\beta = 1.70 \pm 0.49$ SE, 95% CI [0.82, 2.80]). Additionally, males had fewer failed provisioning attempts than females overall ($\beta = -1.45 \pm 0.69$ SE, 95% CI [−2.45, −0.61]; Table S2), and males were only observed to have failed provisioning events during noise trials. There was a weak trend for failed provisioning attempts to increase with brood size, but the precision of the estimated effect was low ($\beta = 1.09 \pm 0.69$ SE, 95% CI [−0.29, 2.66]). Finally, close inspection of the data revealed that many failed provisioning attempts were due to the activity of one female, thus we performed a sensitivity analysis by removing activity from this female from the dataset and rerunning the top model identified with the full dataset. As with the full dataset, the number of failed provisioning attempts was higher during noise trials; however, there was no longer a difference in failed provisioning attempts between male and female parents (Table S1).

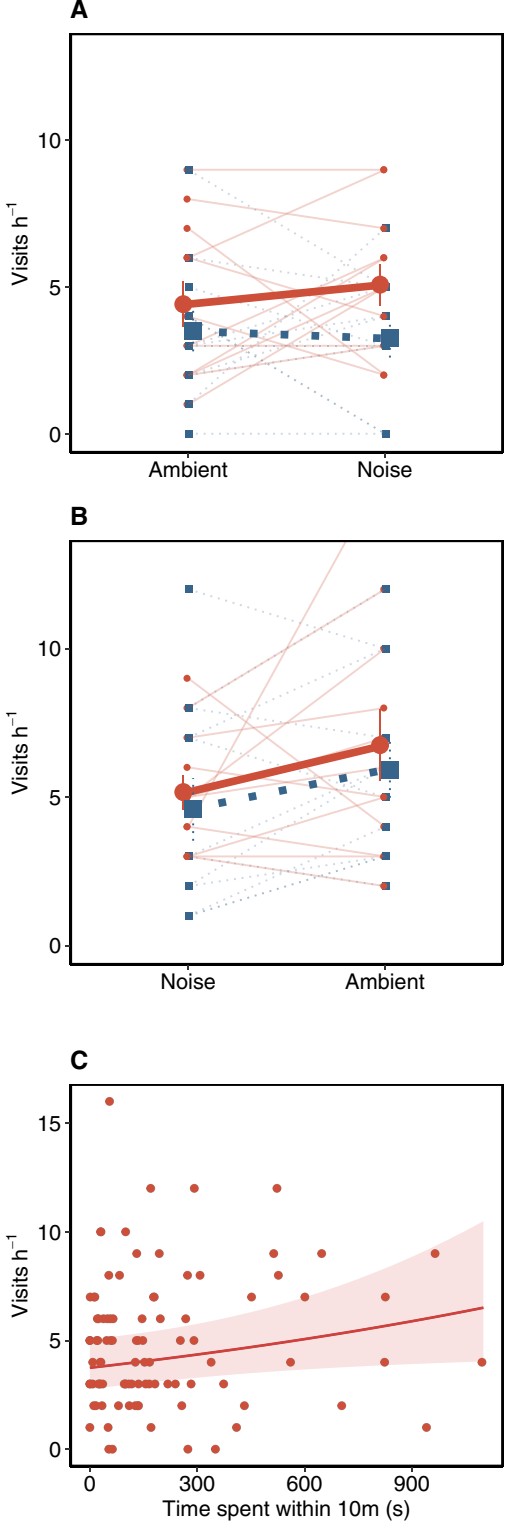

**Figure 2 Western Bluebird provisioning rates the short-term playback experiment.** Western Bluebird provisioning rates under both ambient and noise conditions with a trial order of (A) ambient then noise conditions and (B) noise then ambient conditions. Individual female and male provisioning rates denoted by small red points and blue squares and connected with light solid or dashed lines, respectively.
**Figure 2** (continued)
Large points and squares reflect mean provisioning rates per trial and sex. Vertical error bars on large points and squares reflect ± 1 s.e. Trial order strongly influenced provisioning behavior, such that provisioning rates were higher overall on second trials relative to first trials. (C) Marginal effect of time in seconds parents spent within 10 m on nestling provisioning rates.

**Table 3 Model parameter estimates from the top-ranked model in Table 2 for provisioning behavior of Western Bluebirds in short-term noise exposure trials.**

| Fixed effects | Estimate | SE | 95% CI |
|---|---|---|---|
| (Intercept) | 1.42 | 0.14 | [1.13, 1.70] |
| Treatment noise | 0.22 | 0.19 | [−0.16, 0.59] |
| Trial 2 | 0.53 | 0.19 | [0.16, 0.90] |
| Parent male | −0.24 | 0.09 | [−0.43, −0.05] |
| Time within 10 m | 0.12 | 0.05 | [0.01, 0.22] |
| Treatment noise*Trial 2 | −0.71 | 0.32 | [−1.36, −0.05] |

**Table 4 Model selection table for variables explaining time a Western Bluebird spent within 10 m of the nest box in response to short-term traffic noise playback.**

| Model | K | $AIC_c$ | $\Delta AIC_c$ | weight |
|---|---|---|---|---|
| null | 3 | 359.375 | 0.000 | 0.183 |
| **Treatment** (−), trial (−), **treatment*trial** (+) | 6 | 359.551 | 0.176 | 0.168 |
| Ordinal date (+) | 4 | 360.308 | 0.933 | 0.115 |
| Ordinal date (+), **treatment** (−), trial (−), **treatment*trial** (+) | 7 | 360.479 | 1.105 | 0.106 |
| Brood size (+), **treatment** (−), trial (−), **treatment*trial** (+) | 7 | 360.482 | 1.108 | 0.105 |
| Trial (+) | 4 | 360.779 | 1.404 | 0.091 |
| Brood size (+) | 4 | 360.997 | 1.623 | 0.081 |
| Parent sex (+) | 4 | 361.001 | 1.627 | 0.081 |
| Parent sex (+), **treatment** (−), trial (−), **treatment*trial** (+) | 7 | 361.322 | 1.947 | 0.069 |

**Note:**
Response was natural log transformed after the quantitative adjustment of adding 1 to all values. The ordinal date was center and scaled. +/− shows the direction of the trends. Bolded values show parameters with 95% CIs that do not overlap zero.

**Table 5 Model parameter estimates from 2nd-ranked model in Table 4 for the time a Western Bluebird spent within 10 m of the nest box in short-term noise exposure trials.**

| Fixed effects | Estimate | SE | 95% CI |
|---|---|---|---|
| (Intercept) | 4.86 | 0.34 | [4.17, 5.54] |
| Treatment noise | −0.99 | 0.48 | [−1.95, −0.02] |
| Trial 2 | −0.77 | 0.48 | [−1.73, 0.20] |
| Treatment noise*Trial 2 | 2.02 | 0.79 | [0.40, 3.64] |

**Table 6 Model selection table for variables explaining failed provisioning attempts in response to short-term traffic noise playback for all Western Bluebirds in the study.**

| Model | K | AIC$_c$ | ΔAIC$_c$ | Weight |
|---|---|---|---|---|
| Brood size (+), **parent sex** (−), **treatment** (+) | 6 | 107.52 | 0.00 | 0.14 |
| **Parent sex** (−), **treatment** (+) | 5 | 107.72 | 0.20 | 0.13 |
| **Parent sex** (−), **treatment** (+), trial (−) | 6 | 107.84 | 0.32 | 0.12 |
| Brood size (+), **parent sex** (−), **treatment** (+), trial (−) | 7 | 108.07 | 0.55 | 0.11 |
| Brood size (+), **parent sex** (−), treatment (+), trial (−), treatment*trial (+) | 8 | 108.48 | 0.96 | 0.09 |
| **Parent sex** (−), **treatment** (+), trial (−), treatment*trial (+) | 7 | 108.67 | 1.15 | 0.08 |
| **Parent sex** (−), time within 10 m (+), **treatment** (+) | 6 | 108.70 | 1.18 | 0.08 |
| **Parent sex** (−), time within 10 m (+), **treatment** (+), trial (−) | 7 | 108.93 | 1.41 | 0.07 |
| **Parent sex** (−), ordinal date (−), **treatment** (+) | 6 | 109.05 | 1.53 | 0.07 |
| **Parent sex** (−), ordinal date (−), **treatment** (+), trial (−) | 7 | 109.24 | 1.72 | 0.06 |
| Brood size (+), **parent sex** (−), time within 10 m (+), **treatment** (+) | 7 | 109.35 | 1.83 | 0.06 |
| null | 3 | 117.61 | 10.09 | 0.00 |

Note:
Time within 10 m and ordinal date were centered and scaled. +/− shows the direction of the trends. Bolded values show parameters with 95% CIs that do not overlap zero.

## Continuous noise exposure

Thirteen Western Bluebird nests were included in our analyses spanning sound levels from 58.4 to 82.3 dB. Following model selection, only a single model with sound amplitude as a predictor variable was strongly supported among models explaining provisioning rate and latency to resume provisioning (Table 7). Two models were supported for failed provisioning rate, but the highest-ranked model was the null (Table 7).

Provisioning rates averaged 8.79 (± 4.15 SD) per hour but increased with sound exposure amplitude from approximately 5 visits per hour at amplitudes below 60 dB to more than 14 per hour at nests with amplitudes near 80 dB ($\beta = 0.39 \pm 0.09$, 95% CI [0.19, 0.60], $R^2 = 0.58$; Fig. 3A). Although the latency to resume provisioning after nest approach averaged 443.85 s (± 269.76 SD) across all nests, the latency declined from over 600 s at relatively quiet nests to approximately 250 s at the loudest nests ($\beta = -19.39 \pm 7.96$, 95% CI [−36.90, −1.87], $R^2 = 0.29$; Fig. 3B). Failed provisioning rates averaged 2.53 (± 2.03 SD) events per hour but were unrelated to sound levels and all other predictor variables.

## DISCUSSION

Anthropogenic noise is a globally-widespread sensory pollutant and influences physiology, reproductive success, and behavior (*Barber, Crooks & Fristrup, 2010*; *Shannon et al., 2016*; *Dominoni et al., 2020*; *Francis, Phillips & Barber, 2023*). Because noise has been shown to influence reproductive success and nestling size (*Halfwerk et al., 2011*; *Kleist et al., 2018*; *Injaian, Taff & Patricelli, 2018*; *Ferraro, Le & Francis, 2020*), understanding if and how parental care may change with noise exposure could provide insights into the way(s) in which noise exposure affects nestlings. Our results demonstrate that anthropogenic noise influences Western Bluebird parental care through two separate studies. In short-term experimental trials, experimental noise exposure alone did not influence provisioning

**Table 7 Model selection tables for provisioning rate, failed provisioning rate and latency to resume provisioning models for Western Bluebirds exposed to continuous noise.**

| Response \| Candidate models | K | AIC$_c$ | ΔAIC$_c$ | Weight |
|---|---|---|---|---|
| *Provisioning rate* | | | | |
| **Sound amplitude** (+) | 3 | 69.09 | 0.00 | 1.00 |
| Null | 2 | 78.03 | 8.94 | 0.00 |
| *Failed provisioning rate* | | | | |
| Null | 2 | 59.40 | 0.00 | 0.72 |
| Lay date (+) | 3 | 61.25 | 1.85 | 0.28 |
| *Latency to resume provisioning* | | | | |
| **Sound amplitude** (−) | 3 | 184.44 | 0.00 | 1.00 |
| Null | 2 | 186.58 | 2.14 | 0.00 |

Note:
+/− shows the direction of the trends. Bolded parameters reflect those with effects that have 95% CIs that do not overlap zero.

rates. Instead, we found experimental noise exposure to increase failed provisioning rates and to only influence actual provisioning rates when considering trial order. Specifically, second trials had more provisioning events in ambient sound conditions than in noisy conditions during first or second trials. These results contrast with those from the long-term noise exposure where nests were located in a gradient of continuous noise exposure amplitudes: provisioning rates increased with noise amplitude and failed provisioning attempts were unrelated to noise levels. Additionally, adults returned to the nest more quickly following nest disturbance by human observers with higher noise levels.

Our study found that trial order strongly influenced provisioning behavior during short-term experimental noise exposure. Lower provisioning rates for both ambient sound conditions and traffic noise trials during trial 1 relative to ambient sound conditions in trial 2 may reflect parents' response to the novel stimuli of new equipment in and around their nests. Novel objects can be an acute stressor that increases stress hormones and alter behaviors in several species, including Great Tits (*Parus major*) (*Baugh et al., 2017*) and Tree Swallows (*Tachycineta bicolor*) (*Rivers et al., 2017*). Introducing equipment to the nest area prior to trial 1 may have elicited a stress response that carried over into first trials and overrode any effects of noise (*i.e.*, carryover effect; *O'Connor et al., 2014*). As such, the enduring response to novel objects is not changed with the addition of a second stressor in the form of noise exposure during trial 1 (see *Wilson et al., 2021* for a review of cumulative effects). However, in a lab experiment done on European Starlings (*Sturnus vulgaris*), corticosterone ("CORT") levels typically return to normal basal levels within 60 min following an acute stressor (*Rich & Romero, 2005*). If the stress series is similar for Western Bluebirds, it is possible that CORT levels returned to basal levels prior to trial 2. Under this scenario, for the nests that received the noise treatment in trial 2, traffic noise would have represented a novel stressor that could have re-activated CORT and the associated self-preservation behaviors that are associated with slower provisioning rates. Therefore, trial 2 may be more reflective of behavioral responses due to the noise stimulus, as birds had a longer period to acclimate to equipment and noise was the only new stimulus

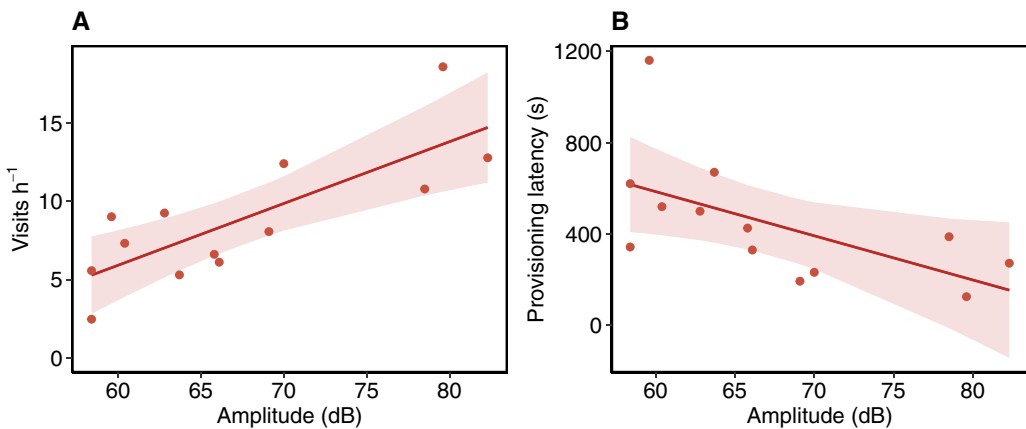

**Figure 3 Changes in Western bluebird parental care behavior along a gradient of continuous noise.** (A) Western Bluebird provisioning rates increased with higher amplitudes of continuous noise. (B) The latency to resume provisioning following nest disturbance decreased with continuous noise amplitude. Ribbons reflect 95% CI of estimated effects.

introduced. This possibility is further supported by the finding that provisioning rates in ambient sound conditions during trial 2 were greater than rates in trial 1 and trial 2 noise-exposed conditions. This finding parallels *Lucass, Eens & Müller*'s *(2016)* work on Great Tits where parental provisioning was lower in experimental noisy conditions compared to control. However, *Lucass, Eens & Müller*'s *(2016)* study did not find an order effect between trials. One possible reason for this may be in part due to mini-speakers hidden within the nesting material of nest boxes rather than placed outside the nest box, eliminating the potential for birds to have a carryover effect from a visual novel stimulus. Future research should consider the effects of the novel stimuli when determining the acclimation period of individual species of birds in experimental trials. Such studies could also consider the possibility that parents might compensate for reduced provisioning by elevating provisioning in second trials above "baseline" rates.

Although not part of our original hypotheses, we found sex-specific differences in total provisioning regardless of noise in short-term experimental trials. Our finding that males provision less in comparison to females regardless of treatment supports previous research showing that females increase provisioning rates in relation to males following the brooding period, beginning around nestling day 5 (*With & Balda, 1990*; *Porras-Reyes et al., 2021*). Although we found no evidence of an interaction between sex and treatment in regard to total provisioning, males in our study had fewer failed provisioning attempts than females when exposed to noise. One possibility for this could be attributed to differences in sexual selection experienced by males and females, as male Western Bluebirds face selection pressures in terms of territorial defense (*Dickinson & Weathers, 1999*; *Naguib et al., 2013*). Given that the principle of allocation can apply to time, males engaged in more territorial defense might not be able to spend as much time provisioning. However, in our sensitivity analysis that excluded one female with many failed provisioning attempts, there was no longer a difference in failed provisioning attempts between males and females. Still, there were more failed attempts in noisy conditions

regardless of the uncertainty on whether males and females differed in their number of failed provisioning attempts. This response could be due to nestlings failing to hear the arrival and call of a parent, which was documented in Tree Swallows by *Leonard & Horn (2012)* in a playback experiment. However, in our study the changes could be due to other possibilities, such as parents failing to hear begging nestlings or changes in hormones among adults or chicks.

Our findings that higher amplitude continuous noise exposure was associated with increased provisioning rates and reduced latency to resume provisioning contrasts with our initial prediction that birds would provision less with increasing sound levels. This finding could, potentially, help explain the complicated pattern of chick size increasing with noise level at lower exposure levels before declining at higher noise levels across three species, including Western Bluebirds (*Kleist et al., 2018*). Despite the potential benefit to nestling growth and condition, why parents increase provisioning and reduce latency to resume provisioning is not obvious. Declines in real or perceived nest predation risk with noise likely explain these relationships. Previous studies in the same study system found that Woodhouse's Scrub-Jays (*Aphelocoma woodhouseii*), which are the key nest predators for songbirds in our study area, avoid areas of high noise, ultimately resulting in reduced nest predation for a variety of nesting species (*Francis, Ortega & Cruz, 2009*). It is possible that Western Bluebirds nesting in noisy areas increased their provisioning rates and decreased their latency time because their perception of nest predation risk was low due to lower densities of nest predators. This trend has also been seen in a multi-species, long-term predator removal study (*Fontaine & Martin, 2006*), which found that parents feed nestlings at a higher rate when predation risk is experimentally reduced. Importantly, however, it is perceptions of risk and not actual risk that drive changes in behavior. In other words, the same changes in behavior could occur when perceived nest predation risk is reduced even when actual nest predation risk is unchanged. For instance, experimental playback of common predator calls to elevate perceived nest predation risk contributed to declines in parental care and reduced reproductive success in Song Sparrows (*Melospiza melodia*) (*Zanette et al., 2011*). Because noise exposure can impair an animal's ability to listen for threats of predators through acoustic masking (*Barber, Crooks & Fristrup, 2010*), a reduction in perceived nest predation risk *via* acoustic masking could also be involved in increased provisioning rates and reduced latency times in areas of high-amplitude noise. However, this possibility conflicts with studies that have evaluated perceptions of adult predation risk in noise *via* foraging-vigilance trade-offs. In lab and field studies, birds and mammals have been shown to increase visual vigilance in noise at the expense of active foraging, presumably due to the reduction in passive acoustic surveillance for threats due to noise (*Quinn et al., 2006*; *Shannon et al., 2014*; *Ware et al., 2015*; *Le et al., 2019*). This increase in vigilance has further been shown in California ground squirrels (*Spermophilus beecheyi*) living under chronic noise exposure, suggesting that not all animals may grow accustomed to high levels of noise over time (*Rabin, Coss & Owings, 2006*). Why responses reflective of perceptions of risk for nest predation in the context of noise may differ from perceptions of adult predation risk is unclear and needs further study, including potential links to hormonal changes during reproduction.

Separately from the perception of nest predation risk and the responses to noise among actual nest predators, hormonal changes in response to acute and chronic stressors may contribute to the difference in provisioning behavior in our short experimental and continuous noise exposure studies, respectively. As discussed above, stress-induced release of CORT due to novel stimuli and noise associated with the short-term noise exposure experiment could suppress provisioning behavior. Additionally, noise has been shown to be a chronic stressor that depresses baseline CORT in systems exposed to noise over long periods of time (*Cyr & Romero, 2007*), including Western Bluebirds exposed to continuous noise (*Kleist et al., 2018*). Lowered baseline CORT may allow birds to maintain behaviors that increase reproductive success, such as provisioning. Therefore, it is possible that the suppression of baseline CORT of birds in this system played a role in the increase of provisioning behaviors and decreased latency time in response to nest disturbance, but more work is needed to explicitly link stress hormone profiles that result from chronic stressors to behaviors that result from a second acute stressor.

Finally, it is worth noting that there are several other differences in the study sites and designs used in our two experiments. For example, nestlings of different ages may have different requirements in terms of parental care. One previous study with Western Bluebirds found that provisioning rates were not influenced by nestling age or brood size (*With & Balda, 1990*). However, another study found that both males and females increased their rate of provisioning as nestlings got older (*Porras-Reyes et al., 2021*). Whether or not provisioning rate changes with nestling age, we did not expect differences in age to change the direction of the effect of noise on provisioning rates. However, *Pandit et al. (2021)* reported that Eastern Bluebird (*Sialia sialis*) parents exposed to simulated anthropogenic noise were reported to provision 1–4 day-old nestlings more frequently than those that were not exposed to noise, but then the pattern reversed when nestlings were older than 11 days post hatching. While the authors suggest the change in their study could be explained by cumulative effects of prolonged noise exposure, the age specific patterns of more frequent provisioning with noise exposure early in the nestling period and less frequently later in the nestling period is opposite the patterns observed in our two studies. The differences observed in our study are likely explained by different physiological responses among parents to short-and continuous noise exposure and the larger community-level changes that occur in landscapes that experience continuous or chronic noise.

## CONCLUSIONS

Overall, our results demonstrate that anthropogenic noise exposure influences parental care behaviors of Western Bluebirds in both the short and long term. There were several differences between our short and continuous noise studies that could potentially explain differences in provisioning behavior in response to noise, including location, habitat context, nestling age, and the type of noise, although comparisons of traffic noise and compressor noise reveal they are quite similar (Fig. S1). It is also possible that the difference in results between the two studies reflect real differences in how animals respond to noise exposure in the short and long term. This possibility clearly needs explicit study.

Nevertheless, our findings could indicate that results from short-term experiments may not accurately reflect how individuals living in real noisy landscapes alter their behavior with noise exposure. This is especially important because much of the research involving the consequences of anthropogenic noise comes from short-term, controlled experiments. Although short-term experiments are key approaches that can control for many confounding variables that complicate observational studies, responses observed on shorter timescales may not adequately capture influential organismal and community-level responses to noise that occur when individuals are exposed to noise continuously. Clarifying if and when behavior differs from experiments and real-world conditions is essential as urbanization expands and changes sensory landscapes throughout the world.

## ACKNOWLEDGEMENTS

We thank Paul Kessler, Maci Lee, Eva Moylan, Isabelle Smits, Sophia Jones, Anjana Kumar, Skyler Meinholz, Ruby Sibul, Kayla Hansen, Makena Keane, Kelley Boland, and Edward Trout for assistance with fieldwork and scoring videos and Emily Taylor and Sean Lema for helpful feedback on earlier drafts of this manuscript.

### Funding

This research was supported by the Natural Sounds and Night Skies Division of the National Park Service (P17AC01178), an American Ornithological Society Research Award and an Animal Behavior Society Student Research Grant. The funders had no role in study design, data collection and analysis, decision to publish, or preparation of the manuscript.

### Grant Disclosures

The following grant information was disclosed by the authors:
Natural Sounds and Night Skies Division of the National Park Service: P17AC01178.
American Ornithological Society Research.
Animal Behavior Society Student Research.

### Competing Interests

The authors declare that they have no competing interests.

### Author Contributions

- Kerstin Ozkan conceived and designed the experiments, performed the experiments, analyzed the data, prepared figures and/or tables, authored or reviewed drafts of the article, and approved the final draft.
- Jordan M. Langley performed the experiments, authored or reviewed drafts of the article, and approved the final draft.
- Jason W. Talbott performed the experiments, authored or reviewed drafts of the article, and approved the final draft.

- Nathan J. Kleist conceived and designed the experiments, performed the experiments, authored or reviewed drafts of the article, and approved the final draft.
- Clinton D. Francis conceived and designed the experiments, analyzed the data, prepared figures and/or tables, authored or reviewed drafts of the article, and approved the final draft.

## Animal Ethics

The following information was supplied relating to ethical approvals (*i.e.*, approving body and any reference numbers):

California Polytechnic State University Institutional Animal Care and Use Committee; University of Colorado Boulder Institutional Animal Care and Use Committee (protocol 2105). Although use of audio recorders or cameras during both the short-term experiments and at nests exposed to continuous noise could potentially impact the privacy of people by inadvertently recording them, this was not an issue in this study. We did not complete short-term trials when people were present and could be inadvertently recorded. In the continuous noise study, our nest boxes are in a very remote area where we have never encountered people near our nests since starting to work in this system in 2005.

## Data Availability

The data and R code are available in the Supplemental Files.

## Supplemental Information

Supplemental information for this article can be found online at http://dx.doi.org/10.7717/peerj.18558#supplemental-information.

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
