# Peer review of "Divergent effects of short-term and continuous anthropogenic noise exposure on Western Bluebird parental care behavior"

_PeerJ, doi:10.7717/peerj.18558_

## Round 0.1 · original submission · Minor Revisions

The reviewers and I agree that this is a sound and publishable study. However, both reviewers have suggestions regarding points that need clarification. I agree with Reviewer 1 that you should be more cautious in attributing the apparently different outcomes of the two experiments to short- vs. long-term effects. It is probably ok to keep the concept in the title, but perhaps give more consideration to alternative explanations in the Discussion and provide a more nuanced statement in the Abstract. It would be appropriate to call for new, well controlled studies on how duration of noise affects parental behavior.

When I was searching for reviewers, I encountered an article by Pandit et al 2021 Behav Ecol 32(4):747-755 which investigates effects of anthropogenic noise on Eastern Bluebirds. You might consider whether its findings are relevant to your study and should be cited. A couple of its citations may also be relevant.

I have provided a pdf using highlights and inserted comments to note minor grammatical errors and statements needing clarification.

Specific comments and suggestions
The Abstract is wordy and unclear in many places. (See my notes on the pdf.) Please completely revise and ask a couple of independent readers to check it for clarity.
L147. Incorrect subhead. You already presented recording methods in the previous paragraph. This section is about analysis. However, I didn’t see anything about analysis of the audio recordings.
L159-165. I could not get a clear picture of the arrangement. There is first a reference to a dichotomous treatment (noisy vs. quiet) and then a gradient but it was not clear where they were placed (the reference to 75 and 175 m is not clear, nor the systematic arrangement). The description implies that there will be 120 data points (10 x 12), but when I look at Fig. 3, I see only 13 points. I am clearly not understanding something about this study design.
L203. Here and elsewhere, you seem to be using the word ‘trial’, which would refer to all treatments provided to a given nest box when I think you mean ‘treatment order’, if I understand correctly. This also applies to the tables.
References. Several of the references have capital letters for most words in article titles, and some are missing italics for species names. In at least one case, page numbers are missing. Please check all carefully, not just the ones I noted.
Fig. 1. I did not find this figure very clear. The caption needs to be much more detailed so that readers understand every element of the figure. The individual components such as speakers need to be described or labelled.Also, the figure refers to the gradual increase in sound over the first 15-min period, which I did not notice being stated in the methods. Using the term ‘broadcast’ for this prior exposure in the exposure does not seem very clear.
Fig. 2. This figure and caption could also be improved. I agree with the reviewer that using the same ordinate scale for both panels would allow reader to more readily see the differences and similarities. Also, it is confusing within the same figure to refer to ambient and control for the same treatment. Some readers may be color blind or read the manuscript as a black-and-white photocopy. It would be helpful to separate males and females by shape of the points in addition to color (and describe in the caption). ‘Error bars reflect SE’ is unclear. If they show plus and minus one SE, please state this explicitly? The last two results describe the patterns. This should be in the Results text and need not be repeated in the caption.
Fig. 3. I think it would help to be clear where zero is on the ordinate. Note here and elsewhere the official SI abbreviation for hours is h.
Tables. Remember, there is an international audience and that not all will have English as their first language: w/in is not a standard abbreviation – spell out. For interactions, asterisk should be between the letters, not a superscript.

Reviewer 1 ·

Basic reporting

This manuscript details two sound experiments looking at the effects of two different anthropogenic noise scenarios on the provisioning behaviour of western bluebirds. The manuscript is well written, and a great addition to the literature on anthropogenic noise.
There are some areas where I have commented that other references are needed that are more appropriate than those given. There are also some minor issues with clarity in some areas.

L18: ‘background’ is not the best word to use here as it is a little ambiguous. Does it mean ambient sounds, anthropogenic sounds?
L19: I understand what you’re trying to say here but logically, if predator cues/alarm calls are masked, you would expect less vigilance (as has been seen in a number of studies).
L20: please specify if anthropogenic noise
L22: missing word/letter here? ‘in the context of a nest’ or ‘in the context of nests’
L24: ‘ambient control’ is still noise – please be specific about what type of noise.
L26: ‘clearly’ feels quite colloquial. If this result is significant, say ‘significantly higher’ instead.
L28: by ‘increases in continuous noise levels’ do you mean increases in amplitude of continuous noise? Might be good to state this it amplitude in the abstract.
L29: what type of disturbance? Maybe say ‘human disturbance/nest check’ to clarify.
L33: maybe instead of ‘appears to conflict’, it ‘appears to be dependent on noise type’ or ‘on temporal factors of noise’

L47: ‘many aspects of’ rather than ‘many aspects on’
L46: I disagree, there is a lot of research looking at how sensory pollutants (e.g. anthropogenic noise and light) affect behaviour. But I do agree that research into which life stages are most affected is lacking.
L51: Importantly, acoustic masking occurs when the frequency of anthropogenic noise overlaps with the frequency of important noises such as an alarm call/mating call. Please clarify this.
L63-65: This sentence doesn’t seem relevant to a study on Western Bluebirds
L80: would be good to say what type of noise (traffic, plane, broadband etc.). Also ‘silent audio track’ seems like an oxymoron.
L82: This is the first time you bring up that noise could affect behaviour via distraction, you have only talked about acoustic masking previously. Would be good to mention this possible mechanism by which noise can affect behaviour further up, with references to support this.
L269: missing bracket.
L303: Reviews by Shannon et al. 2016, Kunc & Schmidt 2019, Kok et al. 2023 etc would be good here as they focus solely on anthropogenic noise, as your study does, rather than all sensory pollutants.
L304: other refs (e.g. ‘Passerine Birds Breeding under Chronic Noise Experience Reduced Fitness’ Schroeder et al. 2012, and ‘Negative impact of traffic noise on avian reproductive success’ Halfwerk et al. 2010.) would be good to show these results are seen beyond your lab and study species.

L306: I think this should be ‘ways in which’ rather than ‘ways by which’
L319: I think ‘lower’ makes more sense than ‘slower’ here

Experimental design

Some areas in the materials and methods need clarification in order to be sure this study is robust.

Materials and methods
L100: add in coordinates of location.
L106: How do you know when ‘completion of a full clutch’ is? Do western bluebirds always lay the same number of eggs? How many is this?
L122: were all roads equally busy? Could be an issue if one road is a highway but another is a local street.
L130: What was the range of received noise amplitude at the nests?
L131: Sound levels were measured during the initial 2 mins of traffic noise tracks post-setup. Does this mean within the 15 minutes of this track being broadcast prior to the beginning of the trial?
L145: Note whether the wrentit call would be expected to have any effect on behaviour.
L148: How did you compare/resolve differences in provisioning behaviour scored from videos after the trial, and behaviour noted by the observer during the trial? Which of these did you use?
L160: in what way are these noisy and quiet sites paired?
L168: did you start recording behaviour immediately after set-up of the video camera, or did you do the same as for the short-term experiment and wait 15mins?
L170: what is the range of time nests were watched for?
176: can you please clarify in the methods section of the short-term experiment whether you noted parental sex?
L211: Is the amount of time spent within 10m of the next box not correlated with amount of provisioning since birds have to be near the nest to provision? If so it should not be included as a predictor in models looking at provisioning attempts.
L261: please check for correlation between time spent within 10, and provisioning, in line with my comment above.

Validity of the findings

Findings are valid, however my only major concern with the manuscript and discussion of your findings is that the point is repeatedly made that based on these results, we shouldn’t rely on results from short-term experiments to tell us about real-world chronic noise effects. While I agree that this is true, I don’t think that you provide robust evidence to say that your ‘results provide a cautionary tale’. Your two experiments are very different; conducted in different states, ten years apart, with different types of noise, of different amplitudes, and on different ages of chicks. Any of these factors (or a combination of them) could lead to your different results. I therefore think you need to be more cautious in your claims.
The continuous noise exposure study also represents a very specific acoustic scenario – compressor engines removed from urbanisation (as you state in lines 186-188, this is a very remote area). Therefore, while this is of course a ‘real-world noisy condition’, I don’t think it can be compared to a short-term experiment in an urban area (near a university campus) and using traffic noise playback.
Also, I think the fact that in your short-term experiment you find a sex effect, but then don’t look at this in your continuous noise experiment (which is understandable because it was conducted 10 years prior) confirms that these two experiments were conducted unrelatedly and then combined in this manuscript.
I think the authors need to be more transparent that you ran experiments investigating the effects of two different noise scenarios on the same species, rather than saying that these results show evidence that short term studies don’t reflect long term studies. This is still a very useful and interesting study.

Additional comments

L252-259: Please be clearer here where you are talking about the effect of trial order alone vs the interaction of treatment with trial order. Currently it is unclear and very difficult to read.
L263: talking about other models when you’re top ranked model is the null seems odd. It’s ok to have a non-significant result.
L302ff: Would be good to mention somewhere in the discussion that ‘treatment alone did not have a strong influence on provisioning behaviour’ in the short term experiment – how do you explain this?
L306: I think this should be ‘ways in which’ rather than ‘ways by which’
L308: state here is noise treatment had an overall effect on provisioning rate, since you are comparing short term results to the continuous experiment provisioning rate and failed provisioning rate results.
L314: specify here that this is only something you looked at in the long-term study, not the short term experiment.
L318ff: Is it possible that this trial order effect is due to birds compensating for a period of reduced provisioning by increasing provisioning in their second trial period? I.e. in response to noise first but also maybe in response to stress due to equipment set up?
L352: I don’t see the link here between males having fewer failed provisioning attempts in noise and males facing selection pressure from territorial and nest defence?
L358: You don’t know from your study the mechanism by which these provisioning attempts failed. It could be that chicks or adults are more stressed in noise, it could be that chicks don’t hear adults approach, it could be that adults aren’t hearing begging calls of chicks that signal hunger. So I don’t think you can conclude that this is similar to findings that there were ‘ a higher number of missed detections in noisy conditions’
L361: Do you mean a higher amplitude of continuous noise exposure?
L366: Are woodhouses scrub-jays a nest predator of other birds?

Reviewer 2 ·

Basic reporting

The manuscript is written clearly and coherently. I have no concerns regarding the layout of the ms. or the professionalism of the language. The 'Introduction' is well-structured and effectively introduces the research aims and questions. The 'Methods' section provides a thorough explanation of the research methodology, which aligns well with the research questions posed. The 'Results' section presents the findings clearly and comprehensively, addressing all the questions raised in the Introduction. The ‘Discussion’ section fully addresses the research questions and refers to all the results obtained. The selection of literature is appropriate and relevant. In my opinion, the manuscript’s structure meets PeerJ standards and adheres to disciplinary norms.
The tables and figures are well-prepared and clear. I have a minor comment regarding some figures, which I will address below.
Thank you for providing the raw data files and R codes used in the statistical analyses, which is in accordance with PeerJ standards. However, I believe it would be beneficial if your supplemental data files included a detailed description of the variables, perhaps in the form of a legend in a separate sheet, to make them more useful for future readers and users.
The field research was carried out with full respect for the welfare of the test animals.

Experimental design

In the paper, the Authors compare the results of two independent field studies investigating the effects of noise on nestling provisioning behavior in western bluebirds. In one study, they experimentally assess the impact of short-term noise exposure (1-hour playback of traffic noise), while in the other, they examine the effects of continuous, chronic noise from a gas well installation. Together, the results demonstrate that noise influences nestling provisioning behavior, but the effects appeared to be opposite in the two study setups. The short-term noise playback experiment showed that noise negatively impacts parameters of nestling feeding behavior, whereas the analysis of continuous noise revealed an increase in nestling provisioning efficiency with higher noise levels. The authors attribute these contrasting results to the different effects of short-term versus continuous (long-term) noise on the birds' parental behavior.
Both experiments reported in this study are very well planned and performed. A methodological issue in the playback experiment - where the focal birds had insufficient time to acclimate to the research apparatus, and the novel, acute sound played near their nests could have caused stress -was constructively addressed in the discussion of the tested hypotheses and the methodology of noise playback studies. To the best of my knowledge, the statistical methods used to analyze the data are appropriate.

Validity of the findings

I have no major concerns regarding the conclusions of this work. Overall, they are well-formulated, closely aligned with the research questions, and remain within the framework established by the results obtained. However, I do have a few comments. I leave it to the authors' decision whether or not they include them in the new version of the ms.
In many parts of your manuscript, you state that you are studying the effects of anthropogenic noise on the parental behavior of birds. However, natural noise sources (acute or chronic), such as wind in windy areas, waterfalls, or thunders, etc., can have similar effects. Perhaps this could be worth addressing in Discussion.
You rightly point out (page 13, lines 343-344) that future research should account for the effects of new stimuli introduced by noise playback, as well as the presence of research apparatus, such as those cameras or microphones placed near birds’ nests. It may be worth specifying more precisely the type of noise involved. In my opinion acute noises, like thunder or the sound of a helicopter, could be effectively used in playback experiments. However, the challenge lies in using continuous, chronic noise, such as road noise. An exception might be the episodic noise from cars or quads rarely traversing wilderness areas.
A faster nestling feeding rate with increasing noise levels (page 14 line 362) could affect nestling growth and condition at fledging, correct? Wouldn't it be worth discussing this?
I generally like this paper and think that it contributes valuable information to our understanding of how affects the parental behavior of birds. It is undoubtedly worth publishing.

Additional comments

I have got two detailed comments:
In the 'Methods' section, you state that 23 nest boxes were used in the short-term noise exposure experiment (page 7 line 112, and page 9 line 190), while in the 'Results' section, you mention 24 nest boxes (page 11 line 242). Please clarify this discrepancy.
Fig. 2. I suggest adjusting both panels of the figure to the same scale on the y-axis. This will make the figure easier to read.

---

## Round 0.2 · accepted · Accept

The authors have responded appropriately to the suggestions of reviewers and editor. I felt it was not necessary to send the manuscript out for a second review and read the revised version carefully myself. I noted a few small errors which are highlighted on the attached pdf.